# Cancer: New Needs, New Models. Is It Time for a Community Oncologist? Another Brick in the Wall

**DOI:** 10.3390/cancers13081919

**Published:** 2021-04-16

**Authors:** Paolo Tralongo, Vittorio Gebbia, Sebastiano Mercadante, Roberto Bordonaro, Francesco Ferraù, Sandro Barni, Alberto Firenze

**Affiliations:** 1Medical Oncology Unit, Hospital Umberto I, RAO, 96100 Siracusa, Italy; 2Section of Medical Oncology, Department of Health Promotion, Mother and Child Care, Internal Medicine and Medical Specialties, University of Palermo, 90100 Palermo, Italy; vittorio.gebbia@gmail.com; 3Medical Oncology Unit, La Maddalena Clinic for Cancer, 90100 Palermo, Italy; 4GSTU Foundation for Cancer Research, 90100 Palermo, Italy; 5Anesthesia and Intensive Care, Pain Relief and Palliative Care Unit, La Maddalena Cancer Center, 90100 Palermo, Italy; terapiadeldolore@lamaddalenanet.it; 6Medical Oncology Unit, ARNAS Garibaldi, 95100 Catania, Italy; oncoct@hotmail.com; 7Medical Oncology Unit, San Vincenzo Hospital, 98039 Taormina, Italy; ferrau@oncologiataormina.it; 8Medical Oncology Unit, ASST Bergamo Ovest, 24047 Treviglio, Italy; sandro.barni@ospedale.treviglio.bg.it; 9Risk Management Unit, AOUP P. Giaccone, University of Palermo, 90100 Palermo, Italy; alberto.firenze@unipa.it

**Keywords:** cancer survivorship, community oncology, clinical governance

## Abstract

**Simple Summary:**

Community care activity in the oncology field does not exist. This unmet need contrasts with the increasing number of patients with a previous diagnosis of cancer.

**Abstract:**

Over the last few decades, thanks to early detection, effective drugs, and personalized treatments, the natural history of cancer has radically changed. Thanks to these advances, we have observed how survival of cancer patients has increased, becoming an ever more important goal in cancer care. Effective clinical governance of survivorship care is essential to ensure a successful transition between active and post-treatment life, identifying optimization of healthcare outcomes and quality of life for patients as the primary objectives. For these reasons, potential intervention models must consider these differences to rationalize the available resources, including economic aspects. In this perspective, analyzing the different models proposed in the literature to manage this type of patients, we focus on the possible role of the so-called “community oncologist”. As a trained health professional, also focused on longevity, he could represent the right management solution in all those “intermediate” clinical conditions that arise between the hospital specialist, frequently overworked, and the general practitioner, often biased by the lack of specific expertise.

## 1. Perspectives

The natural history of cancer has dramatically changed due to early diagnosis, effective drugs, and personalized treatments [1,2,3]. In Italy, in 2021, there will be more than a total of 3.5 million prevalent cancer cases. This number is significantly higher than in the previous decade, in which the patients diagnosed with cancer were 2,637,975, corresponding to a prevalence of 4.6% [4]. Additionally, patients surviving >15 years since diagnosis represent 20% of all prevalent cases in 2010 and could rise to 40% in 2021. The estimated overall trend in the present decade (+3.2% per year) is comparable to that expected in the same period in the USA (+2.8% per year), UK (+3.3%), and Switzerland (+2.5%), highlighting a problem related to the whole world and not only to our national reality [5,6,7]. Overall, the increase of cancer cases over ten years is approximately 37% [5,6,7]. 

Advances in cancer treatment have improved patient survival, making survivorship an increasingly important aspect of cancer care [7,8]. An increase in life expectancy creates new challenges for the oncologist, who must face a new class of problems, which arise from the significant improvements in the cancer patient’s prognosis. All this means that today, investigation oncology may not be just preclinical and clinical research but also clinical assistance modeling. In the real world, most patients are assisted outside research programs and follow pathways suggested by guidelines. In 2020, the European Union promulgated recommendations (Table 1) for psychosocial oncology, rehabilitation, and survivorship research, which emerges the need for and effectiveness of survivorship care models used in various healthcare systems [8].

A large section of the population defined as ‘cancer survivors’ has significant heterogeneity in prognosis and individual characteristics. Survivors are at risk for recurrence of original cancer and at risk for developing other cancers. Therefore, active surveillance and prevention are key components of survivorship programs. A significant proportion of patients have co-morbid conditions that can be exacerbated by the patient’s cancer diagnosis and treatment. Finally, cancer-related lifestyle changes and psychosocial impact can be as devastating as the consequences of the disease itself. They must be addressed, including social relationships, daily activities, sexual habits, work arrangements, health insurance, anxiety, living arrangements, and child care [9,10,11]. These issues are fundamental if we consider that ‘cancer survivors’ include a percentage of patients that we can define as “cured,” which reaches about 27% of the entire population of cancer patients [5,12]. 

Risk stratification is fundamental to optimize the therapeutic and rehabilitative pathways, categorizing patients by disease status [13,14]. About 20% of total survivors have a low probability of disease recurrence or late side effects. On the other hand, numerous subjects present a decidedly higher risk because of organ dysfunction persisting many years after the end of treatment, comorbidity, unhealthy lifestyles, or genetic predisposition. Hence, through risk identification, surveillance must no longer be just aimed at the early diagnosis of disease relapse. However, it should address the prevention of metachronous cancer or late and long-term cancer or treatment side effects and promote correct lifestyles, on top of psychological and social rehabilitation [13,14,15,16].

How survivorship care is structured and delivered is essential to ensure a successful transition between active and post-treatment life and is fundamental to optimize patients’ health outcomes and quality of life. For the above reasons, intervention models must consider these differences to rationalize resources from an economic point of view [17].

While finding ways to meet this growing group of cancer patients’ needs has become a priority, there continues to be an open debate within the oncological community surrounding how to best structure survivorship care [18,19]. 

The workload of cancer centers resulting from the increase in the number of survivors and the financial toxicity of patients linked to travel and uncovered expenses can be reduced thanks to the cooperation and coordination of care between the various health professionals involved in the care of cancer patients through all possible stages of the disease. An effective coordination is essential to achieve this goal in order to guide patients and their families on their journey through the diverse phases of disease. 

In this context, to manage these new requirements, it is essential to consider many different aspects: longevity is a distinct phase of care, specific care plans have to be implemented, continuity between primary care and specialty cancer care has to be promoted, developed, and test the models of care and the guidelines of clinical practice must be defined, as well as the responsibility of clinical research, and, finally, the activity of psychosocial services has to be increased. It is necessary to promote the care team’s training, keeping in mind the longevity perspective to achieve these goals. 

Several models have been proposed to manage this type of patient. The “shared model,” the pathology model, and the multidisciplinary-comprehensive model are the most interesting ones [18,19]. 

### 1.1. The Shared Model

The shared model refers to a patient’s care through the collaboration of two or more specialist doctors with the general practitioner’s intervention. The shared model of care is currently used to manage patients with chronic diseases (diabetes, chronic renal insufficiency, heart disease), and over the past few years, there has been an active scientific debate regarding the applicability of this model to the treatment of cancer patients. This model would seem applicable to the category of “cured” patients [13]. The limitations of this model lie in the difficulty of the general practitioner managing patients with complex problems and the potential challenge of managing non-specialist follow-up in case of unusual outcomes.

### 1.2. The Pathology Model

The “pathology model” involves the activity of a group of disease-specific oncologists dedicated to planning adequate follow-up for all patients’ needs. At a predefined time after completing the therapy, the patients are referred by the nurse case manager to follow-up, and the nurse takes care of re-establishing communication with the general practitioner to start a shared treatment. Reversibility is then followed for some time, related to recurrence risk and any late side effects. The strength of this approach is the relatively low cost, in purely economic and personnel terms. In this care model, it is essential to evaluate the general practitioner’s greater familiarity with most common morbidity (diabetes, dyslipidemia, hypertension), the planning of the follow-up program, the needs of the patients recovered, the reversibility of the patient’s condition (risk of recurrence, late side effects). The latter could reveal an inappropriate nature of the general practitioner. 

### 1.3. The Multidisciplinary-Comprehensive Model

The “multidisciplinary-comprehensive” model is the most complicated program in terms of articulating the centralized interventions. The team comprises doctors trained or experienced in the treatment of cancer survivors, oncology nurses, social workers, psychologists, general practitioners, and a network of consultant specialists such as ophthalmologists, gynecologists, andrologists, cardiologists, nutritionists, neurologists, and other on-demand health professionals. Therefore, this model needs a different set of skills from doctors focused on intensive cancer care. Based on risk, care is provided through teamwork. This model requires, in the beginning, the availability of high economic resources, also because it must also promote research in the area of longevity (prognostic factors, predisposing factors of iatrogenic toxicity, etc.). Coordination and continuous communication with the general practitioner are desirable even if their role is relatively marginal in this organization. 

To date, we cannot say with certainty which is the best model, both in terms of necessary resources and in terms of satisfying all patients’ needs, but, on the other hand, we have full awareness that it is required to: (a) plan an oncological follow-up diversified from “intensive care,” (b) favor the management transfer from the hospital to the community, and (c) identify a coordinator, to avoid a fragmentary and sometimes expensive path for the overlapping of the interventions. Interventions should be personalized and coordinated by team-based continuing care that promotes cross-specialty collaboration after the initial treatment phase. In this contest, identifying the team coordinator is fundamental as it allows for diagnostic-therapeutic interventions to avoid unnecessarily expensive overlapping useless for the patient. Coordination and communication between the different providers are essential for improving these patients’ quality of care and making it personalized [20].

The presence of an integrated care coordinator is mandatory. The coordinator should guide survivorship care, maintain all health professionals informed, and plan the transition from a clinical setting to another at the right time, aiming to reduce patient burden in cancer hospitals and optimize patient management in the chronic stages of the disease (Table 2). The choice of the necessary interventions by the care coordinator should depend on disease stage and phase of disease, the patient’s physical and psychosocial needs (Figure 1 and Figure 2).

In clinical practice, the hospital oncologist has often been the referring physician of any critical situation, placing the general practitioner’s figure in the background. Reviewing the model with the goal of de-hospitalization means rediscovering the territory; this cannot happen without the general practitioner’s involvement. The general practitioner could play an elective role for the diagnosis, persistence of any side effects, and, even more, management of comorbidities, for which he certainly has more experience and more knowledge than the oncologist (Figure 1 and Figure 2). If we consider that about two-thirds of patients with cancer are over the age of 65, we understand how important the involvement of caregivers and the general practitioner is to manage the potential comorbidities that often coexist in these cases [1]. Education and support for caregivers should be an integral part of the action on survivorship care, considering that a significant percentage (15–50%) of them show signs of depression and anxiety [21]. However, numerous studies have highlighted the patients’ preference discrepancies when they have to be followed exclusively by the general practitioner since not all patients share acceptance of this role [22,23,24,25,26].

If we consider these data and the critical issues reported in the evaluation of the individual models, an additional contribution could be to include in the care path the figure of an oncologist who operates in the community. A “community oncologist,” trained and focused on longevity, can intervene in all those “intermediate” clinical conditions that represent an overload for the hospital specialist and, on the other hand, are conditioned by the lack of specific skills on the part of the general practitioner. The acute phase of disease requires hospital care, while a chronic condition with few symptoms can identify the community oncologist as a coordinator who would, in turn, be replaced, in the presence of comorbidities, by the general practitioner.

This pathway would find a rationale for the differences in knowledge and attitudes between primary care practitioners and oncologists [27,28,29,30,31]. Furthermore, studies have shown that general physicians can provide quality cancer care, especially during diagnosis, follow-up, and palliative care phase [32,33].

As in different medical branches such as cardiology, the community oncologist could engage in cancer surveillance, not only disease-oriented. The community oncologist could manage late or long-term iatrogenic side effects could take care of those patients in the condition of chronicity treated with oral (about 30% of patients), subcutaneous drugs, or those at low risk of developmental recovery without comorbidity, could promote the correct lifestyles and plan secondary/tertiary prevention interventions. Health promotion has a priority role since the development of several comorbidities may be favored by pre-existing conditions or may arise due to treatments for the disease, such as cardiac toxicity or metabolic syndrome [34,35,36]. The modifications of lifestyle may have a decisive positive impact on the quality of life and possibly reduce the risk of recurrence and mortality [37].

As is known today, cancer treatments are no longer just infusion therapies. Subcutaneous and oral treatments are becoming more and more frequent [38]. Indeed, the number of welfare models that provide home care treatment are gradually increasing [39]. Moreover, the prognosis has radically changed, and often the disease has a chronic evolution. So many patients find themselves undergoing therapy for a long time in their own homes. The community oncologist could carry out their management to evaluate interventions to prevent, diagnose, and the treatment side effects. The community oncologist should be trained in the survivorship field with masterclasses that highlight the different types of survivors [13] and their specific needs [14]. At the same time, he should organize research on survivorship issues such as descriptive and analytical research, interventional research, examination of survivorship sequelae for unstudied primary cancer sites, surveillance treatments, economic sequelae, economic disparities, family and caregiver issue, instrument development [40].

At the moment, community care activity in the oncology field does not exist. This unmet need contrasts with the increasing number of patients with a previous diagnosis of cancer. This condition could lead hospitals not to support the care needs of a large segment of this population. That is, those chronic or long-term patients at risk of relapse [13,14,15]. Promoting a community activity would prevent the patient from going to the hospital continuously, probably favoring a less traumatic care path.

This activity would be in continuity with what has been planned at the hospital level. The community oncologist must be in the continuity of action with the cancer center. His/her activity would also improve education in the condition of longevity to the patient and his/her family. From a theoretical point of view, the harvest of real-life data on both clinical and management aspects of oncological longevity would enable novel research pathways. The timing for the transition from a cancer hospital to the community oncologist or the general practitioner varies according to the disease’s phase, the characteristics of the disease, the risk of treatment-related complications, and the chance of relapse [41]. 

Undoubtedly, how survival care is structured and delivered is relevant to ensure successful transitions between active and post-treatment phases and optimize patient health outcomes and quality of life. All traditional professionals, oncologists, and general practitioners must be actively involved. The current commitment to ensure the best management of such a program must be focused on collaboration between all health professionals involved in cancer care. 

Such cooperation may allow for an adequate and effective organization of services, an improvement in health education and professional training, a better patient and family education on the condition of long-term living.

The presence of a community oncologist could improve the quality of care and open up further discussion scenarios.

## 2. Declarations

Authors are responsible for correctness of the statements provided in the manuscript.

## Figures and Tables

**Figure 1 cancers-13-01919-f001:**
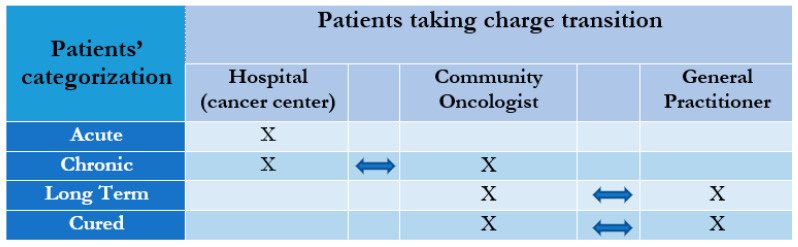
Model of interrelation between patients’ categories and integrated health care.

**Figure 2 cancers-13-01919-f002:**
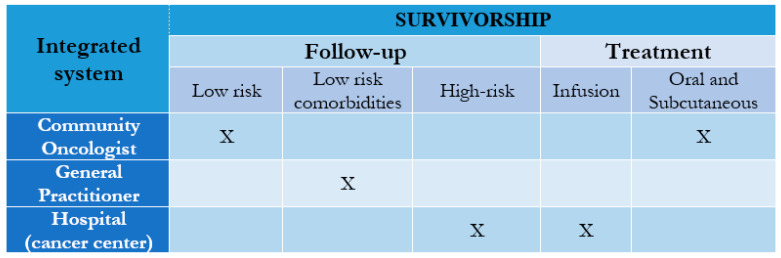
Model of follow-up and treatment according to cancer- and comorbidity-related risk and type of treatment.

**Table 1 cancers-13-01919-t001:** European Union recommendations for psychosocial oncology, rehabilitation, and survivorship research [8].

Support methodological development for assessment of health-related quality of life.
Develop tools to enhance communication with patients and shared decision-making (e.g., increasing patients’ access to their medical records via patient portals, development and testing of decision aids for selecting from available treatments).
Establish international collaboration for developing survivorship-specific patient-reported outcomes in order to monitor the physical and psychosocial health and health-related quality of life of individuals in the post-treatment period. This is a prerequisite for establishing effective programs to address the individual needs of cancer survivors (e.g., return to work, fertility, sexuality, reconstruction surgery, dental health, cognitive functioning, fear of recurrence, etc.).
Develop, test, and implement apps and wearable devices for effective follow-up monitoring and appropriate interventions.
Support research to create a comprehensive overview of the negative consequences of a cancer diagnosis and treatment on physical, mental, and social health in the short and the long term.
Develop prediction models for side effects of treatments.
Support long-term follow-up programs notably for pediatric and young cancer patients to conduct large-scale, longitudinal, observational studies in distinct cohorts of cancer survivors to better understand their problems and needs.
Establish and assess outcomes of guidelines to facilitate return to social health, enable reintegration in the workforce and alleviate financial and legal constraints (e.g., life insurance, mortgage).
Identify health and social inequalities in the cancer survivorship population.
Initiate research on the economic consequences cancer survivors and their relatives are facing. This should include both direct and indirect costs.
Evaluate the need for and effectiveness of survivorship care models used in various healthcare systems.
Conduct research to better understand the causes of differences and discrimination in the survivorship experience between countries and cultures, including financial services such as loans and mortgages.

**Table 2 cancers-13-01919-t002:** Role of the coordinator.

Roles
Keep the doctors involved informed
Guide survivorship care
Plan the transition at the right time

## Data Availability

Not applicable.

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
