# Peer review of "Cancer: New Needs, New Models. Is It Time for a Community Oncologist? Another Brick in the Wall"

_cancers, 2021, doi:10.3390/cancers13081919_

Round 1
Reviewer 1 Report
Survivors should certainly be under the control of many specialistsbecause of the risk of recurrence and the occurrence
of many complications of treatment.
This care is organized differently in each country.
The most important thing is that patients find multi-specialist care.
The authors' proposal is interesting to consider for oncologists.
Author Response
We thank the Referee for approval and we agree with the comments.
Reviewer 2 Report
This is a very innovative manuscript that propose the “community oncology” as a new professional to fill the gap between cancer centres and general practitioners in the management of long-term survivors and analyse in deep the possible models that could be applied to structure survivorship care. The reasons for which this theme is very important are well shown and the literature references are accurate. The existing models of cure for survivors are well discussed. The proposal of the “community oncology” is very well presented.
I think this paper is ready to be published without any modifications except that a very minor one, the sentence:
“In Italy, it is expected that, in 2021, there will be more than a total of 3.5 million cancer cases.”
I suggest “ ...3.5 million prevalent cancer cases”
Compliments to the authors
Author Response
We thank the Referee for the compliments to our paper. The suggested minor text modifications have been made and are highlighted in red.
Reviewer 3 Report
The authors reviewed the various survivorship care models for cancer patients and suggested the presence of a ‘community oncologist’ could improve the quality of care of cancer survivors. As authors described, the number of cancer survivors is increasing because the earlier detection and advances in treatment have improved the survival rates of cancer patients. Therefore, finding an appropriate survivorship care model that can affect the life extension and quality of life in cancer survivors is a very important issue. So, the topic of this article is timely and essential.
However, there are major limitations in this study.
First of all, several articles about this topic were already reported and it’s not specific. And this article does not provide a reasonable and objective evidence for the needs, roles, training methods and expected effects of community oncologists. Moreover, the integrity of the content, such as the overlap of abstract with the text and lack of detailed explanation for the table, is also less faithful.
Unfortunately, it’s not suitable for publication in ‘Cancers’.
Author Response
We thank the Referee for the comments. The abstract has been reworked to avoid overlapping. However, we would like to stress that this is not a study but a perspective paper which propose a continuum care model. As required by the Referee further evidence for ”the needs, roles, training methods and expected effects of community oncologists” have been added in the paper body with related bibliographical references.
In our opinion the needs for a community oncologist are quite clearly reported in our paper (reducing the patients burden to cancer hospital and reducing …. in chronic phases of disease). In our opinion the presence of other articles on care models no way detracts from the solidity of our perspective paper but it adds strength. The specificity is due to the proposal of a new professional figure, namely that of the community oncologist, who is part of a shared assistance model.
This model has been presented to our regional, public health system, government agency which, after accurate evaluation by the strategic and economical bureaus, approved such experimental care system in a Sicilian county under the coordination of one of the author. Therefore we humbly consider the paper suitable for this special issue.
Reviewer 4 Report
Overall, i found the premise of the article intriguing. Although the authors build an excellent case on the basis of survivorship, i find that the issues surrounding the coordination of care are not particularly well-prefaced prior to explaining their concept of a community oncologist. For instance, care coordination is poorly defined and understand yet we all know survivors are falling through gaps in the system and there are challenges in providing regular ongoing care. I want the authors to tease this out more to preface the case. additional grammar. insert comma after "today" line 41, p1. Line 57 - sentence needs reworking as meaning is unclear line 85 - i'd suggest unhealthy lifestyles rather than incorrect 98-101 - the sentence is poorly constructed and fragments are unclear I'd suggest subheadings where "shared model" starts and use section headings for each model line 139 needs "to" before : please address: why should the "The choice of the coordinator should depend on disease stage and phase" shouldn't a "survivorship generalist" coordinate and refer or flag as appropriate to preserve the continuum of care? the work needs a thorough proof read.Author Response
We thank the Referee for the positive comments.
Suggested grammar modifications have been made and highlighted in red as follows:
- comma has been inserted after "today" line 41, p1
- line 57 - sentence has been reworked
- unhealthy substituted for incorrect
- we added subheadings where "shared model" starts and used section headings for each model
- “to" has been added before :
We added information into the text body regarding the coordination of care to introducing and explaining their concept of a community oncologist (highlighted in red)
“The choice of the necessary interventions by the care coordinator should depend on disease stage and phase of disease the patient's physical and psychosocial needs” substituted for "The choice of the coordinator should depend on disease stage and phase". We are sorry for our mistake in structuring the phrase in English.
Round 2
Reviewer 3 Report
Thank you for your efforts.
I applaud the authors for your best efforts.
However, the concept of the 'Community oncologist' in revised version is still abstract and ambiguous.
And, as I already described, it does not contain a resonable and objective evidence for the roles of community oncologist.
So, despite the efforts of the authors, my decision to 'Reject' remains unchanged.
Thank you very much.
This manuscript is a resubmission of an earlier submission. The following is a list of the peer review reports and author responses from that submission.